# Aberrant Responding in Hypothesis Testing: A Threat to Validity or Source of Insight?

**DOI:** 10.3390/bs15030319

**Published:** 2025-03-06

**Authors:** Georgios Sideridis, Mohammed H. Alghamdi

**Affiliations:** 1Boston Children’s Hospital, Harvard Medical School, Boston, MA 02115, USA; 2Department of Self Development Skills, King Saud University, P.O. Box 2454, Riyadh 11451, Saudi Arabia; mhalghamdi@ksu.edu.sa

**Keywords:** aberrant responding, cusp catastrophe model, person-fit indices, U3 index, Guttman errors, student–teacher relations, school safety, nonlinear modeling

## Abstract

Aberrant responding poses a significant challenge in measurement and validity, often distorting well-established relationships between psychological and educational constructs. This study examines how aberrant response patterns influence the relationship between student–teacher relations and students’ perceptions of school safety. Using data from 6617 students from the Saudi Arabia Kingdom from the 2022 Programme for International Student Assessment (PISA), we employed the cusp catastrophe model to evaluate the nonlinear dynamics introduced by aberrant responses, as measured by the U3 person-fit index and the number of Guttman errors. Theoretical and empirical support for the cusp model suggests that aberrance functions as a bifurcation parameter, shifting the relationship between student–teacher relations and perceived school safety from predictable linearity to chaotic instability when exceeding a critical threshold in aberrant responding. Results indicate that both the U3 index and the number of Guttman errors significantly contribute to response distortions, confirming the cusp model’s superiority over traditional linear and logistic alternatives. These findings suggest that ignoring aberrant responding risks misinterpreting data structures, while properly accounting for it through catastrophe models provides a more nuanced understanding of nonlinear system behavior in educational assessment. The study highlights the importance of person-fit statistics in psychometric evaluations and reinforces the predictive utility of nonlinear models in handling response distortions in large-scale assessments.

## 1. Framework for Aberrant Responding in Measurement

The validity of psychological and educational tests depends on the assumption that test respondents respond in a meaningful and predictable manner to test items. Aberrant responding or item-level bias, which generally refers to patterns of response that diverge from the expected, however, is a major threat to measurement ([21]). Aberrant responses take the form of random responding disengagement ([42]), response sets ([40]), guessing ([1]), and cheating, all of which introduce large systematic measurement error ([19]; [10]). Such aberrant responses not only cause inconsistencies at the individual level but can also disrupt the (usually) strong associations among psychological constructs, resulting in all types of erroneous conclusions regarding large-scale educational assessment. Therefore, the aim of the current study was to examine the role of aberrant response patterns in the relationship between student–teacher relations and students’ perceived school safety, employing person-fit statistics and nonlinear catastrophe modeling.

Aberrant responding has typically been measured using person-fit indices which measure how much a person’s response pattern differs from what we would expect based on item response theory (IRT), parametric or non-parametric ([36]; [23]). Some of the most popular person-fit statistics include the U3 index ([36]) and the number of Guttman errors ([29]). As a local index, the U3 index is a residual-based measure that detects local inconsistencies between observed and expected data ([19]), and, thus, is particularly sensitive to careless responding, inattention, or fatigue in certain sections of an assessment. In contrast, Guttman errors measure global response misfit as the extent of an individual’s violation of the anticipated ordering of item difficulties ([29]). Both indices have been shown to be practical tools for detecting outliers in data across fields ranging from educational measurement to clinical psychology and large-scale assessment research ([20]; [19]).

Many traditional statistical methods rely on the assumption of linear relationships between multiple variables, but the introduction of aberrant responding into an assessment setting can create a disruption in linearity, which would be best captured by more advanced modeling techniques ([36]). In contrast to random noise, which oftentimes attenuates relationships by adding stochastic variance, aberrant responding adds systematic measurement error by introducing multiple modes to several groups. For example, a group of students who are sitting next to a window with no blinds may be adversely affected by the sun, adjacent noises, and the heat; thus, their performance may take on a negative trajectory. On the other hand, participants without these conditions may perform more optimally in relation to their true state. Thus, aberrance may be manifested with distinct subgroups who perform similarly in relation to achievement with the total distribution exhibiting multimodality ([37]).

One such approach is catastrophe theory, which captures small continuous changes in predictor variables that can ultimately result in abrupt, discontinuous changes in an outcome variable ([34]). The cusp catastrophe model, a special case of catastrophe modeling, is especially useful for investigating discontinuous change in behavior as a function of two interacting control parameters, an asymmetry parameter and a bifurcation parameter ([13]; [11]). Specifically, in the context of the present study, we hypothesize that student–teacher relations play the role of the asymmetry parameter, imparting a stable, linear positive relationship with students’ feelings of school safety, whereas aberrant responding (i.e., number of Guttman errors and the U3 index) acts as the bifurcation parameters, causing instability and unpredictability in the system. We hypothesize that as the magnitude of personal levels of aberrance rises beyond a critical tipping point, the otherwise linear relationship between student–teacher relations becomes unpredictable, leading to various modalities in behavior, reflecting a chaotic structure ([37]; [7]). A more detailed description of the cusp catastrophe model is presented in Section 4.

## 2. Studying Student–Teacher Relations and School Safety

Interpersonal relations between students and teachers are widely cited as one of the most important predictors for students’ school safety perceptions, with positive interactions being associated with academic performance, emotional status, and a decrease in school-based violence overall ([8]; [39]). Theories that explain school climate suggest that students’ sense of safety is determined by institutional factors (e.g., school policies; disciplinary structure) as well as interpersonal factors (e.g., supportive relationships with teachers and peers) ([3]). Previous studies have indicated that caring student–teacher relations serve as a protective factor against bullying, classroom conflict, and victimization and can enhance the emotional security of the school ([12]; [32]). Nevertheless, these well-established relationships are based on the premise that students provide valid and interpretable responses in self-report measures.

In large-scale assessments, it has been suspected that factors reflecting the validity of these responses are operative. At the very basic level, international assessments such as Programme for International Student Assessment (PISA (PISA) screen for one such pattern, involving straightlining responses. For that purpose, a sizable percentage of responders per country is deleted prior to sharing these datasets. This percentage can reach up to 6% of the total sample, which at times may involve more than 600 participants. However, other factors including survey fatigue, disengagement and motivation may also be operative in ways that systematically compromise the validity of measurements in large-scale surveys (e.g., PISA), when the length of the assessment or low stakes lead to low efforts in the completion of the assessment ([26]). If high levels of aberrant responding are undetected, then inferences about associations between variables such as between student–teacher relations and school safety may be inaccurate and invalid. Not accounting for aberrant responding, therefore, stands to underestimate or overestimate the actual effect of student–teacher relations on perceived safety, which could inform interventions designed to create safer school environments.

## 3. Importance of the Present Study

Based on the above, the present study adopts a different role for the presence of aberrant responding utilizing nonlinear modeling. Specifically, we employed the cusp catastrophe model to investigate how aberrant responding moderates the relationship between student–teacher relations and students’ feelings of school safety. Using data from the 2022 PISA cohort, we test the hypothesis that a well-known, valid relationship between student–teacher relations and school safety becomes distorted when aberrant responding exceeds a critical threshold. Specifically, we propose the following:Aberrant responding (measured by U3 and the number of Guttman errors) acts as a bifurcation parameter, leading to discontinuous and unpredictable changes in school safety perceptions when present in large amounts.The cusp model will provide a better fit to the data compared to traditional linear and logistic models, demonstrating the need for nonlinear modeling in studying measurement artifacts.The relationship between student–teacher relations and school safety will be significantly moderated by aberrant responding, leading to differential effects depending on the level of aberrance in the data.

The present study advances current knowledge in three important ways. It offers a new application of catastrophe theory in educational measurement, showing how aberrant response patterns can interfere with the interpretation of inferences made from large-scale survey data. Second, it reinforces the validity of the person-fit indices (U3 and Guttman errors) as necessary, performs prerequisite analyses in psychometrics, and confirms their predictive value for the detection of disengaged or careless responding ([21]). Third, it points to the importance of using nonlinear models like the cusp catastrophe model that may capture complex interactions that are lost when using traditional statistical techniques ([14]). The integration of linear modeling and complex systems theory in this study represents an advance on existing methods for dealing with response distortion and offers a robust approach to the interpretation of student-reported school climate and safety data.

## 4. Major Research Hypothesis and Analytical Model

The major thesis of the present study is that the presence of aberrant responding is so detrimental for measurement purposes in that originally salient relationships between constructs are masked when aberrance is present. There is unfortunately evidence that specifically in international databases such as PISA, large numbers of respondents are classified as aberrant or unconstructive for measurement purposes. Specifically, in PISA, it is acknowledged that up to 6% of responses per country are flagged and subsequently excluded prior to public release due to straightlining, i.e., only a single form of aberrant responding ([26]). In low-stakes survey conditions in PISA, instances of careless responding have also been observed (e.g., [35]). In the present study, we hypothesized that the presence of aberrant responding distorts an existing and well-known relationship between two constructs, to the effect that the relationship no longer holds. These thoughts align with the cusp catastrophe model in which the presence of a control parameter (termed bifurcation) distorts a linear relationship between two variables so that behavior becomes unpredictable and chaotic, taking on various modalities (multimodal). Figure 1 describes the theses of the cusp catastrophe model.

As shown in Figure 1, the cusp catastrophe model offers a nonlinear view for examining how two control parameters (student/teacher relations and aberrant responding) interact and alter the relationship between an independent variable (student/teacher relations) and an outcome variable (feelings of school safety). There are two control parameters: asymmetry, which introduces a linear relationship between student/teacher relations and feelings of safety, and a bifurcation parameter (aberrant responding), which captures instability, unpredictability, and chaos in the system. The dependent variable, “feeling safe at school”, occupies the behavior plane, where the relationship shifts between linearity and nonlinearity depending on the interplay between the two control parameters. As shown in the figure, the behavior of the system transitions from a smooth, linear trajectory to a nonlinear, catastrophic trajectory under certain parameter conditions. Specifically, the null point and inaccessible zone denote areas where traditional linear modeling fails to explain abrupt changes in the outcome. In this study, we test the hypothesis that aberrant responding, as indexed by U3, acts as a bifurcation parameter, amplifying or dampening the relationship between student–teacher relations and students’ feelings of safety. The transition from linear to nonlinear relations and the potential moderating role of aberrant responding, which exceeds a critical threshold, result in unpredictability that advances our understanding of complex dynamics in educational settings.

## 5. Goal of the Present Study

The goal of the present study was to test the hypothesis that a valid and well-known relationship between student–teacher relations and a school’s perceived safe climate is distorted in the presence of aberrant responding by several participants. Specifically, we hypothesize that when aberrance is measured using person-fit indices, it exceeds a critical cutoff point, and the well-known relationship no longer holds and becomes chaotic. These theses are tested using the cusp catastrophe model that is suited for evaluating such complex, nonlinear relationships.

## 6. Method

### 6.1. Participants and Procedures

Data came from 6617 students from Saudi Arabia as per the 2022 cohort of the PISA database ([26]). There were 5982 10th graders and 623 11th graders (data were missing for 12 students). There were 3471 females (52.5%) and 3146 males (47.5%). The spoken language at home was Arabic for 6102 students (92.3%) and other than Arabic for 511 students (7.7%). The language of testing and assessments was Arabic for 6336 students (95.8%) and English for 281 students (4.2%). PISA adopts a two-stage stratified random sampling design ([26]) to achieve representativeness and reduce selection bias. In the first stage, established by Bayesian sample size calculation, schools were randomly selected with a probability proportionate to a measure of their size, stratified by key characteristics, including geographic region, school type, and socioeconomic indicators where applicable. In the 2nd stage, students in sampled schools were selected at random to participate, resulting in a representative cross-section of 15-year-olds within and across educational systems. Additionally, PISA utilizes a replication approach, normally balanced repeated replication (BRR) or jackknife repeated replication (JRR) to produce estimates of standard errors and obtain robust variance estimates. Technical reports outlining procedures and methods as well as the ethical procedures involved are described here (https://www.oecd.org/pisa/publications/pisa-2022-results.htm, accessed 2 February 2025). Data may be accessed at https://www.oecd.org/en/data/datasets/pisa-2022-database.html, accessed 2 February 2025.

The present study analyses publicly available data from the Programme for International Student Assessment (PISA) conducted by the Organization for Economic Co-operation and Development (OECD). Informed consent was obtained from students and their guardians before participating. Ethical review and approval for PISA procedures were conducted by the OECD and national governing bodies; the OECD ensures adherence to ethical principles by reference to international standards and specifically the Declaration of Helsinki. All personally identifiable information was removed from the data, and the data were anonymized before they were made publicly available to protect the confidentiality of respondents. The present study involves a secondary data analysis and hence does not include any interaction with human participants. Therefore, no institutional review board (IRB) approval was necessary. Additional information on PISA’s ethics procedures, informed consent processes, and data security can be found in PISA’s official technical ([26]).

### 6.2. Measures

All measures were derived from the PISA 2022 cohort and are described below.

#### 6.2.1. Student–Teacher Relations

This scale comprises 8 items completed by students related to the teacher’s care for them. Sample items are “The teachers at my school are respectful towards me” or “If I walked into my classes upset, my teachers would be concerned about me”. The scaling involves 4 Likert-type options anchored between strongly agree and strongly disagree. The marginal reliability was estimated at 0.85. Scaled scores using weighted likelihood estimation were used as reported in OECD.

#### 6.2.2. Feelings of Safety

This scale comprises four items employing a 4-point Likert-type scaling system ranging from strongly disagree to strongly agree. Sample items are “I feel safe on my way to school” or “I feel safe in my classrooms at school”. The marginal reliability of the scale was 0.78. Estimated theta scores using weighted likelihood estimation (WLE) comprised the dependent variable.

#### 6.2.3. Aberrant Responding

Two non-parametric person-fit indicators were utilized to evaluate aberrant responding, namely, the U3 and Guttman error indices. Their use and functionality have been confirmed in the literature (e.g., [19]) and have been used because they highlight aberrance both at specific parts of the measure (i.e., locally) and on the instrument as a whole (global assessment). The U3 index is sensitive to localized aberrance when, for a given level of the latent trait (i.e., an unobservable variable comprised by a number of measured items), several items appear to behave in unexpected ways. It is a residual-based statistic and very sensitive to deciphering behaviors related to inattention, fatigue, and other contextual factors ([19]; [21]). Based on the simulation by [19] ([19]), U3 had the highest predictive ability among 36 person-fit aberrance indices using ROC analyses (see also [9]). On the other hand, the number of Guttman errors is a likelihood-based index using the ideal Guttman pattern in that levels of responding are expected to be higher for items at low latent trait importance and lower for items that saliently define the latent trait. For example, in a depression measure, an item about not wanting to socialize may be common and be endorsed by most participants, whereas an item about self-harm may be very uncommon and be selected by very few participants. The deviation from a perfectly ordered structure in that high-ability persons are successful in items below their skill level creates the ideal Guttman scale. Thus, such inconsistencies are termed Guttman errors (see Table 1).

### 6.3. Data Analyses

#### 6.3.1. Evaluating Person Aberrance Levels

The U3 statistic ([36]) evaluates the extent to which an individual’s response pattern deviates from the expected item response pattern using a non-parametric IRT model and more specifically Mokken’s Monotone Homegeneity Model (MHM, [23]; [24]), thus posing no prerequisites to parametric IRT modeling in that a specific functional form is required (e.g., logistic or normal ogive). It is calculated as(1)$$U_{3}=\frac{S^{+}}{S^{+}+S^{-}}$$
where $S^{+}$ is the sum of positive deviations between the observed score and the expected score and $S^{-}$ is the sum of negative deviations between the observed score and the expected score. Consequently, given an item response function (IRF) $P_{i}(\theta )$, which represents the probability of a correct response to item i given ability levels θ, the expected item score is(2)$$E\left(X_{i}\right)=P_{i}(\theta )$$

Subsequently, the deviations from the expected score for a person *i* on item *X*, with n responses are calculated as shown below:(3)$$\begin{array}{ll} & S^{+}=\sum_{i=1}^{n}\;\;max\left(X_{i}-E\left(X_{i}\right),0\right)\\ & S^{-}=\sum_{i=1}^{n}\;\;max\left(E\left(X_{i}\right)-X_{i},0\right) \end{array}$$

If a person responds consistently according to the model, then $S^{+}$ and $S^{-}$ should be balanced, resulting in a U3 value close to 0.5. Extreme values near 0 or 1 indicate aberrant responding, as the pattern deviates substantially from the expected probabilities.

The Guttman error index estimates the number of violations of the expected item ordering and, similar to U3, employs a non-parametric version of IRT which reflects a deterministic hierarchical item ordering. If an instrument follows a Guttman scale, a higher ability level should lead to a higher probability of correctly answering more difficult items. A Guttman error occurs when a person answers a difficult item correctly but misses an easier one. In calculating the number of Guttman errors (G), we compare the responses of an individual across all pairs of items (i,j), where item i is easier than item j based on item difficulty parameter estimates.(4)$$G=\sum_{i<j}\;I\left(X_{i}<X_{j}\right)$$
where $X_{i}$ is the response to item i (1 for correct, 0 for incorrect, $X_{j}$ is the response to item j and $I\left(X_{i}<X_{j}\right)$ is an indicator function that equals 1 if a Guttman error occurs, and 0 otherwise. Perfect Guttman patterns result in scores equal to zero. Large values indicate the number of errors and inconsistencies, likely pointing to careless responding. Both indices were estimated using the Perfit package ([31]), in the R environment (R 4.4.2, [27]).

#### 6.3.2. Cusp Catastrophe Model: Description

The cusp catastrophe model has recently been popularized in the social sciences with the goal of examining scenario where shuttle changes in a parameter are associated with drastic and dramatic changes in outcome variables. The cusp catastrophe model employs a potential function, *f*(y; a, b), for a single dependent variable y given linear and nonlinear parameters a and b:*f* (y; a, b) = **a**y + 1/2**b**y^2^ − 1/4y^4^(5)

The phenomenon under study is termed the “catastrophe set” which evaluates outcomes inside the parameter space of coordinates (a, b). As shown in Figure 1, regarding the “b” terms (bifurcation variables), in our instance, aberrant responding is at low levels and changes in school feelings of safety are similarly expected to vary linearly with changes in student/teacher relations. However, when aberrance levels in the sample increase beyond some critical acceptable level (reflecting random measurement error), feelings of school safety oscillate between two behavioral modes, low and high, reaching a state of unpredictability.

Available analytical means to capture the theses of the cusp catastrophe model involve [7]’s ([7]) methodology and its implementation in R (cusp package; [11]), the polynomial regression model proposed by [14] ([14]), and the modifications made by [6] ([6]) to Cobb’s methodology. We choose [7]’s ([7]) model given its wide application using the cusp package in R. Support of the model is provided by confirming statistically significant tests in the contribution of the asymmetry and bifurcation parameters. Additionally, inferential statistical tests need to show a preference for the cusp model over its nested counterpart, the linear model. Furthermore, some observations need to fall within the bifurcation area with minimum recommendations for 10% of the total sample size ([11]; [15]) and there must be evidence pointing to bimodality within the bifurcation area and multimodality elsewhere for the outcome variable ([30]). Last, the pseudo-R-squared statistic should be superior in the cusp model over the linear and logistic models ([16]). Amongst criteria, the logistic comparison model is very important as it models an s-shaped relationship, similar to the one of the cusp catastrophe but without the bifurcation structure ([5]). Thus, the major difference between the logistic and cusp models is that the former models smooth, s-shaped transitions ([11]; [25]).

## 7. Results

### 7.1. Prerequisite Analyses of the Cusp Catastrophe Model

One of the important assumptions of the cusp catastrophe is that the dependent variable must have multiple modes. We employed several tests and viewed the distribution of the outcome variable to visually examine its shape and number of modes. As shown in Figure 2, the “feelings of safety” latent variable was not normally distributed. To test the presence of multiple modes, we employed [17] ([17]) dip test that examines the difference between the empirical distribution and a theoretical unimodal distribution ([17]). Results indicated that the alternative hypothesis of the distribution being “at least bimodal” was supported [Hartigan’s D = 0.131, *p* < 0.001]. Further evidence was provided by examining the number of modes using the multimode package in R ([2]). Based on the results from that test, the number of modes was greater than 1 and less than 7, agreeing with the visual inspection that 5 modes are clearly visible as tested with the mode tree test ([22]) (see Figure 2).

### 7.2. Examining the Role of Aberrant Responding to the Prediction of School Safety from Student–Teacher Relations

Table 2 displays the global model fit results with the bifurcation term being aberrant responding either using the U3 index or the number of Guttman errors. Using any of the two aberrant responding indicators resulted in identical results in that model fit was superior for the cusp model over both the logistic and linear models (for U3) based on both information criteria values, and the R-square statistic, which increased from 9.5% for the logistic model to 31.9% for the cusp model. Furthermore, given the nested nature of the linear and cusp models, their chi-square difference test again supported the superiority of the latter [χ^2^(2) = 1046.00, *p* < 0.001]. Similarly, for the model where the splitting factor was the number of Guttman errors, model fit was superior for the cusp model over the linear model [χ^2^(2) = 953.00, *p* < 0.001] as well as the logistic model. The amount of explained variance in school safety increased from 8% in the linear model to 29.1% for the cusp model.

Table 3 displays parameter estimates after fitting the cusp model separately for the U3 and Guttman error models. As shown in the table, in the upper panel, using U3 as the splitting factor, all intercept and slope terms of the cusp model were significant. Student–teacher relationships were a positive predictor of feelings of school safety (b_S-T_ = 0.274, *p* < 0.001). Furthermore, the bifurcation term for U3 was significant and positive for safety (b_U3_ = 1.696, *p* < 0.001), pointing to nonlinearity. Specifically, as aberrant responding increases beyond some critical point that no longer signals random measurement error, the relationship between student–teacher relations and school safety is altered and distorted in a way that it can no longer be estimated using linear means. Thus, the increase in aberrant responding beyond some critical level invalidates an existing positive relationship, which eventually becomes unpredictable and chaotic. The same effects were observed for the number of Guttman errors, with the bifurcation term being again significant and positive (b_Guttman Errors_ = 0.019, *p* < 0.001). Further evidence for the cusp model’s preference is shown in Figure 3, with multimodal distributions at various areas across the response surface and bimodality within the bifurcation area (bottom right figure). The number of observations within the bifurcation area exceeded the minimum cutoff of 10%, being 18.34% when U3 was the bifurcation term and 18.35% when the number of Guttman errors was the bifurcation term. Figure 4 displays observations with darker colors indicating those that are closer to the upper surface and light colors indicating those that are closer to the lower surface. Last, Figure 5 displays the observations as they oscillate between upper and lower surfaces and within the bifurcation area again fitting the expectations of the cusp catastrophe model.

## 8. Discussion

The present study sought to examine the impact of aberrant responding on the well-established relationship between student–teacher relations and perceptions of school safety, using the cusp catastrophe model. Our results confirm that aberrant responding acts as a bifurcation parameter, introducing instability and nonlinear disruptions to what is traditionally viewed as a linear association between student–teacher relations and feelings of school climate. When aberrant response levels (as measured by U3 and Guttman errors) remain below a critical threshold, student–teacher relationships are linearly related to students’ perceived school safety. However, as aberrant responding increases beyond some critical level, this relationship is manifested with multiple modes in the outcome variable, pointing to a chaotic relationship.

Comparing model fit statistics, the cusp catastrophe model consistently outperformed both linear and logistic models, explaining 31.9% of the variance (U3) and 29.1% (Guttman errors) in school safety, whereas traditional logistic models explained only 9.5%. This finding aligns with previous studies suggesting that person-fit statistics capture systematic response distortions that can obscure meaningful relationships in survey-based educational assessments ([21]; [19]). The presence of bimodal and multimodal distributions in the outcome variable further supports the notion that traditional parametric models may fail to account for the complex dynamics introduced by aberrant responding ([38]). These findings provide compelling evidence that aberrant responding is not merely a source of random measurement error that is both acceptable and expected, but a means to fundamentally alter the structure of existing relationships. By modeling aberrant response behaviors explicitly, we gain a deeper, more nuanced understanding of how response distortions affect large-scale assessment data and how best to mitigate their effects in educational research and policymaking.

Our findings align with prior studies emphasizing the impact of person-fit statistics in psychometric evaluations, reinforcing the importance of the U3 and Guttman errors indices as reliable indicators of disengagement, inattention, and careless responding ([20]; [29]). Consistent with research on catastrophe theory in psychology ([14]; [11]), our results confirm that nonlinear disruptions emerge when aberrant responding exceeds a certain threshold, a phenomenon also observed in studies of sudden attitude shifts ([38]) and cognitive development transitions ([37]).

The observed bifurcation effect aligns with theories of school climate and school safety, which suggest that positive student–teacher interactions represent a critical protective factor against school violence and classroom disruptions ([8]; [39]). However, prior research has largely assumed a stable, linear relationship between these factors, neglecting the possibility that response distortions may introduce systematic bias into estimates of school safety perceptions. By demonstrating that systematic measurement error can fundamentally alter well-established relationships, this study challenges prevailing assumptions in school climate research and highlights the need for incorporating nonlinear modeling techniques in future analyses.

Our findings also have important implications for large-scale assessments such as PISA. Research suggests that low-stakes testing environments (such as PISA) are particularly susceptible to disengagement and aberrant responding ([26]). Our study reinforces this concern by showing that high levels of aberrance systematically distort the interpretation of key school climate variables, which may affect cross-national comparisons and educational policy decisions.

### 8.1. Limitations

Despite the advantages of this study, there are several limitations that need to be considered. First, as a secondary data analysis on PISA 2022, we had limited capacity to control for test-taking conditions and respondent motivation. Although PISA uses strict survey administration and quality control procedures ([26]), the specific cognitive and affective drivers of implausible responding are not well understood using quantitative-only means. Future work should explore qualitative methodologies, for instance interviews, think-aloud procedures, or response-time methods, to shed light on the psychological mechanisms underlying aberrant behavior. Second, even though U3 and the number of Guttman errors are well supported for their predictive validity, they represent only a fragment of aberrant response patterns. Many more indicators need to be tested so that they correspond to additional types of response distortions such as extreme response style, acquiescence, or more. Other analytical models such as latent class analysis, and machine learning classification models might offer a richer framework for identifying aberrance ([19]; [21]). This raises two issues: First, the cusp catastrophe model we developed captured only the nonlinear effects of aberrance. Second, its generalizability beyond PISA’s international survey framework remains an open question. Since PISA is an assessment with low-stakes consequences, it is possible that aberrant responding would take different forms in high-stakes contexts—for example, at university entrance examinations or employment aptitude tests—where motivational factors may play a more prominent role. Finally, our study focused solely on student–teacher relations and school safety, but it is possible that additional school climate constructs including peer relationships, school leadership, and teacher well-being may moderate the relationship between aberrant responding and other outcomes.

### 8.2. Implications of the Findings for Educational Policy

These findings have important implications for educational policy, assessment design, and psychometric practices. First, policymakers and test developers could begin to incorporate real-time aberrant response detection mechanisms into large-scale assessments, which would mitigate the impact that disengaged or inattentive responding has on data quality. Considering these considerable distortions to school climate estimates we find introduced by aberrance, we recommend organizations such as OECD, NAEP, TIMSS and PISA move toward adaptive testing strategies in which flagged respondents are offered engagement checks or subjected to different test administration protocols. Second, the findings underline the importance of multi-method approaches in evaluating school safety and student well-being—self-report measures alone may inflate or understate true perceptions when aberrance levels are high. School administrators and policymakers are well advised to complement student-reported climate measures with teacher and administrator reports, direct behavioral observations, and incident-based records to triangulate more accurate estimates of school safety. Finally, the nonlinear modeling methods explored, including the cusp catastrophe model, should be applied to other aspects of educational and psychological measurement that may involve discontinuous changes and threshold effects. In traditional methods, linear regression models, it is assumed that the rate of change is constant; hence, changes in students’ behaviors, for example, loss of academic motivation, shift in engagement or sudden burning episodes, can be abrupt, discontinuous and difficult to model accurately. Incorporation of nonlinear modeling frameworks for intervention programs and large-scale survey analyses may prove more accurate and actionable, providing educators with the ability to identify early signs of withdrawal early on. Lastly, international comparative education researchers need to be more cautious with cross-national comparisons of PISA data because differences in aberrant response across cultures and types of education systems could produce misleading policy recommendations. We call for an awareness of person-fit indicators in advising and interpreting practice-relevant rankings for inter-country educational assessments, because all inter-group benchmarking practices need to rely on valid inferences.

### 8.3. Future Directions

Based on this study, there are a few potential directions for future research. First, follow-up research should investigate alternative methods for identifying and categorizing aberrant responding, such as latent transition models, machine learning algorithms, and unsupervised clustering methods. These approaches could yield more fine-grained taxonomies of aberrance, distinguishing random guessing from strategic faking and fatigue from disengagement. Second, experimental studies should identify the causal mechanisms that produce aberrant responding by directly manipulating test-taking conditions such as time pressure, incentives for motivation, and changes in cognitive load to understand how such changes in the testing process produce distortions in response ([41]). Such research could help distinguish motivationally driven aberrance (e.g., low test stakes leading to disengagement) from cognitively driven aberrance (e.g., poor item readability, item difficulty mismatches). Third, greater cross-cultural research should be performed to understand how the differences in culture and educational systems may influence levels and impact of aberrant responding. Some societies place more value on standardized testing and academic competition than others, leading to reduced disengagement but increased strategic responding, compared with attention to test-taking in other contexts that may promote higher levels of test anxiety, with more erratic response behaviors ([28]). Further research should explore whether aberrant responding varies systematically by country, socioeconomic status, or test-taking motivation. Fourth, further investigation should examine whether the nonlinear disruptions resulting from aberrance can be generalized beyond school climate variables to other areas of educational measurement, including, but not limited to, student achievement trajectories or academic resilience, or cognitive growth modeling. Fifth, future research should focus on intervention strategies for reducing aberrant responding in low-stakes and high-stakes testing contexts, including the design of engagement-enhancing tests (i.e., gamified tests, adaptive tests, and personalized feedback mechanisms that can keep respondents engaged throughout a test). Last, aberrant responding can be extended to the study of extreme response styles ([4]) or sets ([33]) or the use of unbiased aberrance detection methods ([18]).

## Figures and Tables

**Figure 1 behavsci-15-00319-f001:**
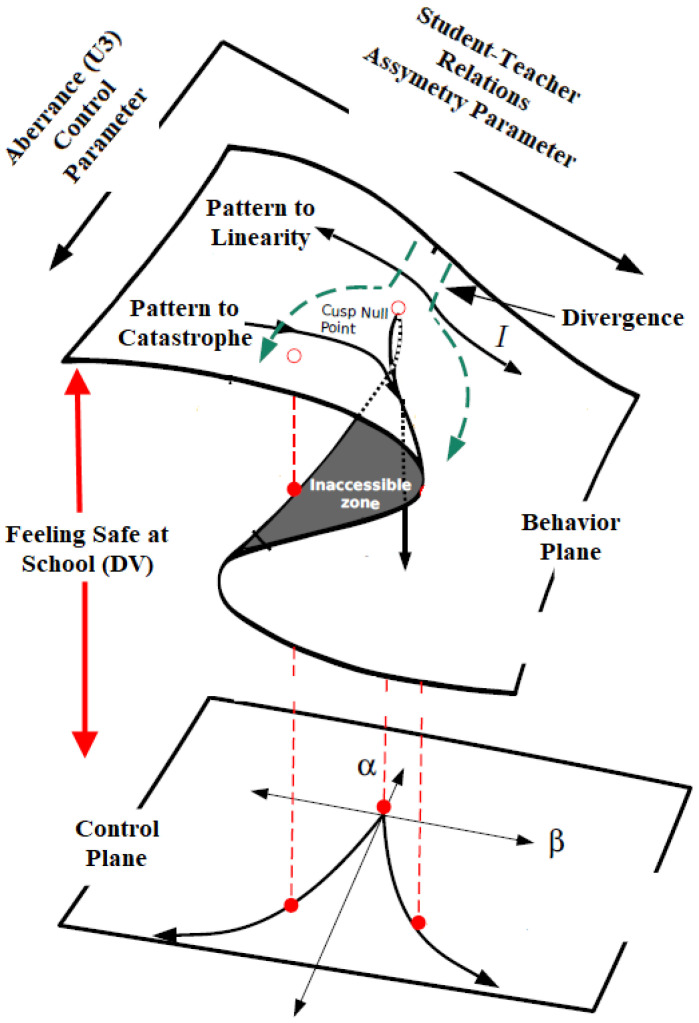
Thom’s theoretical formulation of the cusp catastrophe model.

**Figure 2 behavsci-15-00319-f002:**
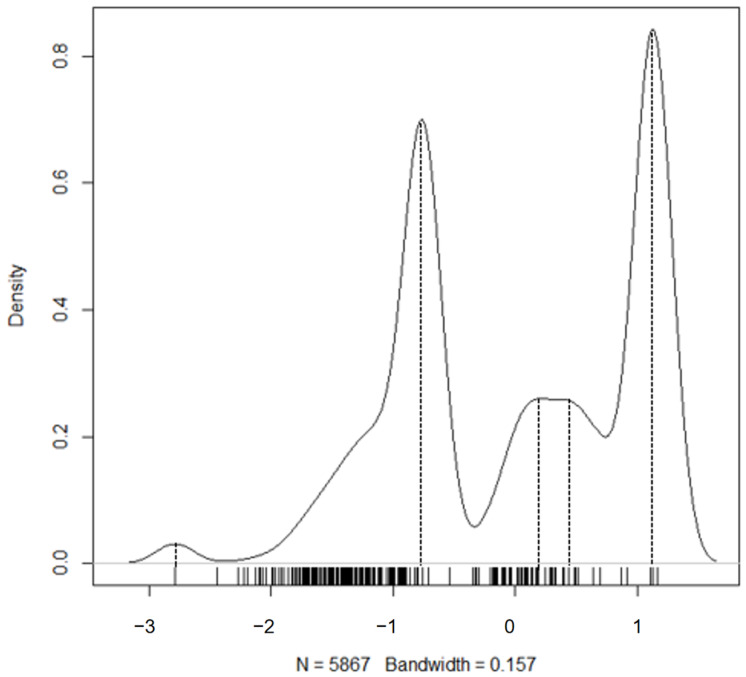
Density of outcome variable with dashed lines pointing to identified modes.

**Figure 3 behavsci-15-00319-f003:**
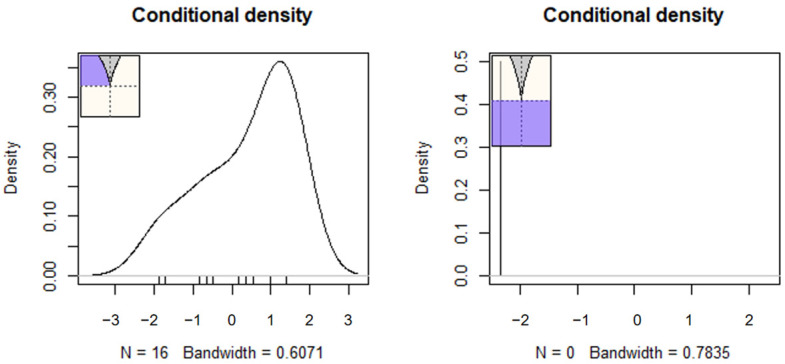
Densities per area of the lower surface for the model using U3 as the bifurcation term. The respective estimates when the bifurcation term was the number of Guttman errors are shown in Appendix A. The colored area indicates the densities within and outside the parts of the lower surface.

**Figure 4 behavsci-15-00319-f004:**
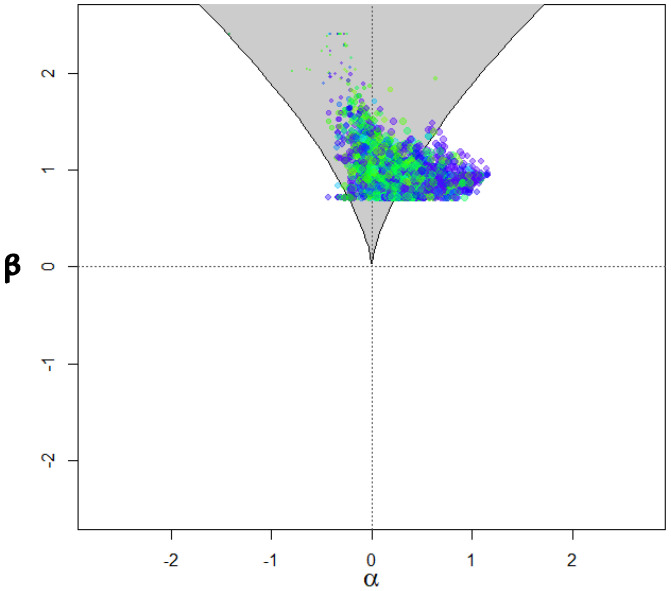
Observations in the lower surface within and outside the bifurcation area for the cusp model with the U3 index. Darker colors point to observations that are present closer to the upper surface and the opposite.

**Figure 5 behavsci-15-00319-f005:**
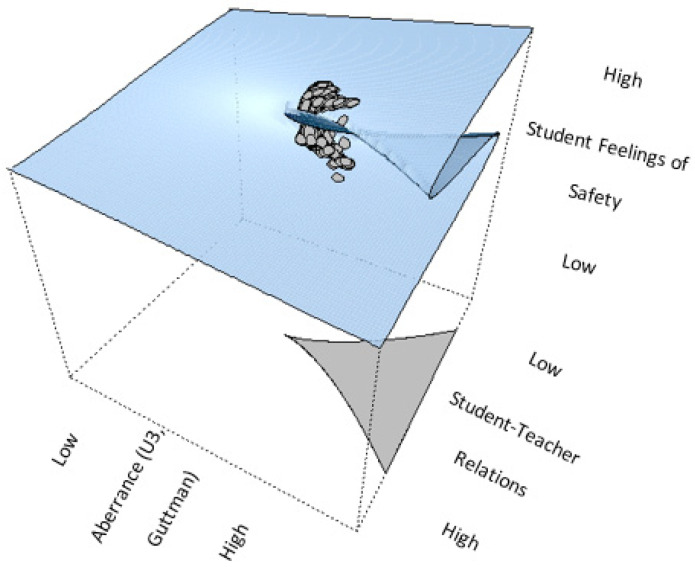
Oscillation of observations between upper and lower surfaces with U3 as the splitting factor.

**Table 1 behavsci-15-00319-t001:** Description of person-fit indices with a local and global focus.

Goal	U3	Guttman Errors
Target	Local inconsistencies	Global response misfit
Aberrance Detected	Erratic responses to specific items	Systematic patterns or biases
Estimation Method	Residual-based	Likelihood-based
Sensitivity in Detecting	Inattention/fatigue/contextual factors	Systematic biases/carelessness/guessing/extreme response style

**Table 2 behavsci-15-00319-t002:** Global model fit using the two aberrant response indices.

Model	R-Squared	logLik	Npar	AIC	AICc	BIC
U3						
Linear	0.079	−8020.634	4	16,049.270	16,049.270	16,075.980
Logistic	0.095	−7970.330	5	15,950.660	15,950.670	15,984.050
Cusp	0.319	−7497.490	6	15,006.980	15,006.990	15,047.040
Guttman Errors						
Linear	0.080	−8018.734	4	16,045.470	16,045.470	16,072.180
Logistic	0.095	−7970.781	5	15,951.560	15,951.570	15,984.950
Cusp	0.291	−7512.091	6	15,036.180	15,036.200	15,076.240

**Table 3 behavsci-15-00319-t003:** Estimated cusp catastrophe models using the U3 and Guttman error aberrance indices.

Coefficients:	Estimate	S.E.	z-Value	*p*-Value	LCI_0_._025_	UCI_0_._975_
U3 Bifurcation Term Model						
a[(Intercept)]	0.285 ***	0.020	14.010	<0.001	0.245	0.324
a[RELATST]	0.274 ***	0.013	20.360	<0.001	0.248	0.301
b[(Intercept)]	0.714 ***	0.050	14.360	<0.001	0.616	0.811
b[U3poly]	1.696 ***	0.169	10.050	<0.001	1.365	2.027
w[(Intercept)]	0.280 ***	0.012	22.980	<0.001	0.256	0.304
w[FEELSAFE]	1.009 ***	0.010	102.320	<0.001	0.990	1.028
Guttman Errors Bifurcation Term Model						
a[(Intercept)]	0.289 ***	0.020	14.707	<0.001	0.250	0.327
a[RELATST]	0.279 ***	0.014	20.542	<0.001	0.252	0.306
b[(Intercept)]	0.691 ***	0.056	12.249	<0.001	0.580	0.801
b[Gpoly]	0.019 ***	0.002	7.479	<0.001	0.014	0.023
w[(Intercept)]	0.287 ***	0.012	23.254	<0.001	0.263	0.311
w[FEELSAFE]	1.004 ***	0.010	101.744	<0.001	0.985	1.024

*** *p* < 0.001.

## Data Availability

Data are available from the official study of PISA 2022.

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
