# Peer review of "Aberrant Responding in Hypothesis Testing: A Threat to Validity or Source of Insight?"

_behavsci, 2025, doi:10.3390/bs15030319_

Round 1

Reviewer 1 Report

Comments and Suggestions for Authors

Comments to authors

Generally speaking, the paper is well written.  However, I did find some typos (e.g., P. 1, line 37 “causes” should be “cause”).  Therefore, I would encourage the authors to give the manuscript a careful edit.

  1. 2, line 45 It appears that the first paragraph on page 2 is identical to the last paragraph on page 1.
  2. 2, line 60 The authors assert that aberrant responding can lead to nonlinear relationships between scale scores and other variables. However, they don’t discuss (a) what types of nonlinearities might be induced by aberrant responding and (b) why this might be the case.  It is intuitive, in my opinion, that aberrant responding would distort relationships among variables, particularly attenuating them due to an additional unaccounted for random component in the item response process.  It seems less clear why nonlinearities would be generated.  I’m not questioning the authors’ assertion, but am asking that they provide a stronger justification for it and an explanation of possible processes that might cause it.
  3. 2, line 63 The authors refer to the cusp catastrophe model, the U3 statistic, and Guttman errors, but don’t define any of them until much later in the paper. This organization of the manuscript is probably fine, but the authors should tell the reader that these terms will all be described in the Methods section.  Alternatively, the authors could move the description of these statistics and the model to an earlier position in the manuscript and mitigate the confusion that some readers unfamiliar with these terms might have.
  4. 3, line 135 In the statement of the major research hypothesis, the authors argue that aberrant responding is a major problem in responses to PISA survey questions. However, they do not provide evidence supporting this assertion.  Is there prior research to support widespread aberrant responding?  More support for this hypothesis should be provided, in my opinion.
  5. 6, line 230 I am not sure that some readers will be familiar with a number of the technical terms used by the authors in this paragraph. For example, IRT models are not described, nor is the concept fo the latent trait.  The example appearing in line 233 is somewhat helpful in this regard, but I think that a bit more description of what IRT models are, what a latent trait is, and how the two concepts are related to one another would be very helpful.  I don’t think that this additional discussion needs to be highly technical, but some additional information would help readers unfamiliar with these ideas.
  6. 6, line 244 Which IRT model was used in the current study?
  7. 6, line 255 Please be sure to define X in equation 3.
  8. 7, line 257 An important requirement for using U3 is that the correct IRT model be fit to the data. The authors need to make this clear in their description and they need to spend a little time discussing how they assessed model fit (and which IRT model they used).
  9. 7, lines 293-299 The authors provide a nice discussion about how the relative and absolute fit of the cusp catastrophe model are assessed. However, they don’t provide any citations for these, which I think should be done.
  10. 8, Results section I found the results section to be very well written and clear. I think that it will be quite helpful for researchers interested in using this model.
  11. 13, Future directions In reading the manuscript, I wondered about the possibility of using one of the IRT based approaches to modeling various aberrant responding patterns (e.g., Bolt & Newton, 2011; Jim & Wang, 2014; Thissen-Roe & Thissen, 2013). It seems to me that incorporating indices of misfit due to aberrant response styles would be an interesting extension of the current work.

Author Response

Thank you for your comments, please see attached file for an item by item response.

Reviewer 2 Report

Comments and Suggestions for Authors

The study presents a novel application of the cusp catastrophe model in educational assessment and provides valuable insights into aberrant response patterns. The manuscript is well-structured and addresses an important issue in psychometric evaluations. However, a few comments below regarding the statistical approaches would help improve the overall quality.

  1. Provide additional justification for choosing the Cusp Catastrophe Model over other nonlinear models in the first section or the beginning of the manuscript.
  2. What criteria were used to determine critical thresholds beyond which aberrant responding leads to chaotic behavior?
  3. The authors mention a minimum cutoff of 10% for observations in the bifurcation area. What is the justification for this threshold, and how does this choice impact model sensitivity?
  4. Provide the data source link.
  5. The R package cusp was cited, but no details regarding hyperparameter tuning or implementation specifics are provided. It is recommended that a publicly available code or sample code with proper code comments be provided.

Author Response

Thank you for your constructive comments, an item by item response is attached.

Round 2

Reviewer 1 Report

Comments and Suggestions for Authors

I've read through the authors' responses to my comments and am satisfied with them. 

Reviewer 2 Report

Comments and Suggestions for Authors

N/A